Changing health compliance through message repetition based on the extended parallel process model in the COVID-19 pandemic

http://orcid.org/0000-0002-6050-656X Yang Jingwen 1 yangjingwen0921@gmail.com
http://orcid.org/0000-0002-9461-3558 Wu Xue 1
http://orcid.org/0000-0002-5496-3748 Sasaki Kyoshiro 2
http://orcid.org/0000-0003-1431-568X Yamada Yuki 3
1 Graduate School of Human-Environment Studies, Kyushu University , Fukuoka , Japan
2 Faculty of Informatics, Kansai University , Takatsuki , Japan
3 Faculty of Arts and Science, Kyushu University , Fukuoka , Japan
Patton Bob
Electronic publication date: 2020 Nov 2
Publication date: 2020
Volume: 8
Electronic Location ID: e10318
Received 2020 Jul 14; Accepted 2020 Oct 16
Copyright: © 2020 Yang et al.
Copyright year: 2020
Copyright holder: Yang et al.
License: This is an open access article distributed under the terms of the Creative Commons Attribution License, which permits unrestricted use, distribution, reproduction and adaptation in any medium and for any purpose provided that it is properly attributed. For attribution, the original author(s), title, publication source (PeerJ) and either DOI or URL of the article must be cited.
License URL: https://creativecommons.org/licenses/by/4.0/

Keywords: COVID-19, Health communication, Pandemic response, Hand hygiene, Infection prevention, Persuasiveness

Funding: JSPS KAKENHI JP19K14482, JP16H03079, JP17H00875, JP18K12015 and JP20H04581 This research is supported by JSPS KAKENHI JP19K14482 to Kyoshiro Sasaki, and JP16H03079, JP17H00875, JP18K12015, and JP20H04581 to Yuki Yamada. The funders had no role in study design, data collection and analysis, decision to publish, or preparation of the manuscript.

==============================
When people are confronted with health proposals during the coronavirus disease 2019 (COVID-19) pandemic, it has been suggested that fear of COVID-19 can serve protective functions and ensure public health compliance. However, health proposal repetition and its perceived efficacy also influence the behavior intention toward the proposal, which has not yet been confirmed in the COVID-19 context. The present study aims to examine whether the extended parallel process model (EPPM) can be generalized to a naturalistic context like the COVID-19 pandemic. Additionally, we will explore how repetition of a health proposal is involved with the EPPM. In this study, two groups of participants are exposed to the same health proposal related to COVID-19, where one group is exposed once and another group twice. They then fill out a questionnaire consisting of items concerning behavior intention and adapted from the Risk Behavior Diagnosis Scale. Structural equation modeling will be used to determine the multivariate associations between the variables. We predict that repetition of the health proposal will associate with response efficacy (i.e., a belief about the effectiveness of the health proposal in deterring the threat) and perceived susceptibility (i.e., a belief about the risk of experiencing the threat). It is also predicted that following the EPPM, behavior intention will associate with both perceived efficacy of the health proposal, which will underlie response efficacy, and perceived threat of COVID-19, which will underlie perceived susceptibility. We will discuss the process, based on the model, where health message repetition affects behavior intention during the COVID-19 pandemic.

What is the Main Question Being Addressed in Your Study?

At the time of writing, it has been 7 months since the outbreak of the COVID-19 pandemic. During this period, while a heavy loss of both life and economy has been caused, some countries and regions have achieved staged success in the fight against this disease. There is no doubt that public compliance with effective health proposals plays a crucial role in achieving this success.

It is suggested that functional fear, which is explained as one of the negative emotions serving protective functions in certain contexts, has promoted public health compliance during the COVID-19 pandemic (Harper et al., 2020). Nevertheless, this needs to be explored further, especially considering previous studies on health communication and the features of information dissemination in real life. The main question being addressed in our study is to examine how people’s health compliance intention is influenced by various factors in the COVID-19 context in an exhaustive way.

It is known that fear has an impact on health compliance. In fact, fear appealing communication is considered an effective way to promote health campaigns and has been widely investigated for promoting health awareness related to various topics, including smoking (Leventhal & Watts, 1966), alcohol use (Wolburg, 2001; Moscato et al., 2001), AIDS (Treise & Weigold, 2001) and so forth. After several initial studies, inconsistent results indicated that a simple monotonic function of fear may not be expected in persuasive health communication. Specifically, despite a large number of studies indicating the positive main effect of fear on persuasion (Leventhal & Watts, 1966; Dabbs & Leventhal, 1966; Leventhal, Singer & Jones, 1965), relatively few report the negative main effect of fear (Janis & Feshbach, 1953; Janis & Terwilliger, 1962). Later research confirmed that the effects of fear appeal interact with various source variables, message variables, and receiver variables, and thus cannot be described easily (Miller & Hewgill, 1966).

One of the most recent and prevalent theories on fear appealing communication is the extended parallel process model (EPPM: Witte, 1992, 1994), which is based on former frameworks including the Parallel Response Model (Leventhal, 1970) and protection motivation theory (Rogers, 1975). In the EPPM, there are four main factors, which influence the prediction of certain communication outcomes: perceived susceptibility and severity composing perceived threat, self-efficacy and response efficacy composing perceived efficacy. Perceived susceptibility refers to a belief about the risk of experiencing a threat, whereas severity refers to a belief about the magnitude of the threat. On the other hand, self-efficacy is defined as a belief about the ability to perform a recommended proposal to avert the threat; response efficacy is a belief about the effectiveness of the recommended proposal in deterring the threat (Witte, 1996). Concretely, when both perceived threat and perceived efficacy are high, people are most likely to engage in a danger control process, which means conforming to the recommended health proposal. Nevertheless, when perceived threat and perceived efficacy are high and low, respectively, people turn to a fear control process leading to coping responses that reduce fear and danger control responses. Meta-analyses on the results of fear appeal research confirm the validity of the EPPM (Floyd, Prentice-Dunn & Rogers, 2000; Peters, Ruiter & Kok, 2013).

Currently, we are exposed to health proposals concerning COVID-19, in all likelihood, more than once in our daily lives, which makes message repetition an essential factor to be taken into consideration. Although research concerning stimulus repetition first emerged in the 1960s (Zajonc, 1968), there was no study on message repetition in persuasive communication until 10 years later. It is suggested that under moderate repetition (less than three times), agreement toward persuasive messages rises (Cacioppo & Petty, 1979). The rationale is explained as follows: Scrutiny is reinforced through a moderate level of message repetition, which enhances the understanding of message content and the merits advocated by it, consequently improving supportive attitudes toward the message. Later research revealed that the mechanism mentioned above is applicable only when arguments in the message are perceived as strong, and when the issue is of high personal relevance (Cacioppo & Petty, 1989; Claypool et al., 2004).

In health communication on COVID-19, we assume that the interpretation of the content of a certain health proposal message may be related to factors in the EPPM. Concretely, response efficacy and perceived susceptibility may be perceived when reading a health proposal message. If the content of a health proposal is supported, high response efficacy will be found, resulting in a decrease in perceived susceptibility. Given the connection between the EPPM and message repetition in persuasive health communication and the actual state we have been through when confronted with health proposals during the COVID-19 pandemic, we built an integrative model to investigate the factors and their associations concerning an individual’s behavior intention to conduct effective health proposals to prevent the infection. In the model, we will first focus on the influence of message repetition on the response efficacy of a certain health proposal and the perceived susceptibility of COVID-19. We will then elaborate on the change in behavior intention toward the proposal due to the variation in perceived efficacy and perceived threat. Furthermore, we will confirm whether the underlying roles of perceived efficacy and perceived threat hold true in the COVID-19 pandemic. The present study is unique and necessary in several aspects.

One, considering that COVID-19 is a real-life and ongoing public health emergency, some of its properties are fixed and thus cannot be manipulated. Take perceived threat for example: In previous research, it was always considered together with fear because without extra fear-arousing materials (e.g., explanation of a certain disease in text or video), the perceived threat may not be notable enough to act as an independent variable. Unlike other unfamiliar diseases (e.g., melanoma), even though the degree to which one is influenced by the pandemic may vary, COVID-19 should be one of immediate health threats in several countries and regions including Japan, where the present study will be conducted. Considering that fear-arousing information is no longer needed in the message, we are interested in exploring how perceived threat alone for the COVID-19 affects behavior intention when there is no intentional manipulation of fear arousal.

Two, there is still no research to test a model that combines the EPPM with message repetition. Although studies on similar topics have been conducted, the results have not been analyzed in an integrated way (Skilbeck, Tulips & Ley, 1977; Treise & Weigold, 2001; Shi & Smith, 2016). To be specific, when supportive arguments toward the content of a recommended proposal are enhanced due to moderate repetition, it can also be interpreted as the change in response efficacy and perceived susceptibility in the EPPM. Nevertheless, this connection between two research topics has rarely been made clear, causing difficulty in making an exact prediction of the results.

Three, it remains essential to confirm the validity of EPPM’s established construction, namely the sub-dimensions of perceived efficacy and perceived threat, to enhance compliance with public health guidelines in the COVID-19 pandemic. We assume better health compliance is due to higher perceived efficacy, but what does perceived efficacy mean? By learning more about its two indicators, self-efficacy and response efficacy, we can better understand why people choose to conform or not to certain health proposals. For instance, even though physical distancing is considered efficient in preventing infection (high response efficacy), the difficulty in conducting it may vary between people based on their socializing needs (divergent self-efficacy), which results in different levels of health compliance. Similarly, if we are aware that perceived threat is indicated by perceived susceptibility and severity, we may know which properties to emphasize in education on COVID-19 to boost public health compliance.

In summary, the present study is of comprehensive significance in revealing the mechanism in the purview of public compliance with effective health proposals in the COVID-19 pandemic.

Describe the key Independent and Dependent Variable(s), Specifying how they will be Measured

We will conduct a two-wave survey for two groups of participants: One is a no-repetition group and the other is a repetition group. In the first wave, two groups will first answer the same dummy questionnaire of the Need for Cognition Scale (Kouyama & Fujihara, 1991). At the end of the questionnaire, only the repetition group will be exposed to the target health proposal message, which is written in Japanese as:

The English translation of the message is as follows: “While alcohol-based hand sanitizer is useful for preventing COVID-19 infection, the effect will be discounted if not enough amount is used. Therefore, it is recommended to press the pump to the bottom every time to get enough amount of hand sanitizer.” We selected this health proposal message according to a pilot study to examine how much the message is known and agreed on. As per the results, the knowledge rate of the message was low (40.8%). The attitudes to the message were measured using a 7-point scale (ranging from 1 = strongly disagree and 7 = strongly agree). The attitude to the message was considerably favorable (i.e., the scores were significantly higher than 4 on the scale, M = 5.35, SD = 1.29, t (200) = 5.52, p < 0.001, Cohen’s dz = 1.047). Considering the definition by Cacioppo & Petty (1989) that a strong argument refers to the one toward which favorable thoughts are generated predominantly, the above message is appropriate in the present study in two ways. One is that as mentioned above (i.e., “What is the Main Question Being Addressed in your Study?”). The supportive arguments on a message can be improved through moderate repetition only when the message is believed to be a strong one; therefore, our manipulation on message repetition will be meaningful. The other reason is that since the scores on favorable thought are moderately high, there is still space for supportive arguments to increase, preventing ceiling effects.

The second wave will be conducted 24–72 h after the first wave. The interval range is set because we cannot control the precise timepoint at which participants answer the second questionnaire even if we invite them to do it on time. In the second wave, both groups of participants will be exposed to the same message, which is identical to the message shown to the repetition group in the first wave. They will then fill out the same questionnaire containing 13 items (Table 1). Twelve items are adapted from the Risk Behavior Diagnosis Scale (Witte, 1996), and 1 item enquires behavior intention toward the target health proposal. All items will be scored on a 7-point scale ranging from 1 (strongly disagree) to 7 (strongly agree). In the preliminary experiment, which had the same design as the main experiment (more details in Supplemental 1), we checked the convergent validity and discriminant validity for items 1–12. The results of our calculations are as follows: (1) average variance extracted (AVE) > 0.5; and (2) square root of AVE > inter-construct correlations are both met, confirming the validity of the items (Hair et al., 2010). The scores of the 13 items in Table 1 will be treated as the key dependent variables.

Table 1 Items included in the questionnaire (items 1–12 based on The Risk Behavior Diagnosis Scale, item 13 on behavior intention).

Self-Efficacy
(1-strongly disagree, 7-strongly agree)	
1	I am able to perform the underlined proposal to prevent the infection of COVID-19	
2	It is easy to perform the underlined proposal to prevent the infection of COVID-19	
3	I can perform the underlined proposal to prevent the infection of COVID-19	
Response Efficacy
(1-strongly disagree, 7-strongly agree)	
4	Performing the underlined proposal prevents the infection of COVID-19	
5	Performing the underlined proposal works in deterring COVID-19	
6	Performing the underlined proposal is effective in getting rid of COVID-19	
Perceived Susceptibility
(1-strongly disagree, 7-strongly agree)	
7	I am at risk of being infected with COVID-19	
8	It is possible that I will get infected with COVID-19	
9	I am susceptible to COVID-19 infection	
Severity
(1-strongly disagree, 7-strongly agree)	
10	COVID-19 is a serious threat	
11	COVID-19 is harmful	
12	COVID-19 is a severe threat	
Behavior Intention
(1-strongly disagree, 7-strongly agree)	
13	In the future, when sanitizing my hands with alcohol-based hand sanitizer, I will press the pump slowly to the bottom to get a sufficient amount	

We aim to test the model (Fig. 1) which combines the EPPM with message repetition, further exploring the factors that associate with behavior intention. As described above, there will be two conditions with various frequencies of exposure to the message. Each participant will be assigned to one of the conditions. Thus, repetition, as a binary variable in the model, will be adopted as the key independent variable.

Figure 1 An integrated model of health compliance intention in the COVID-19 pandemic.

What are Your Hypotheses?

The hypotheses of this research are:Hypothesis 1: Perceived efficacy has a positive effect on self-efficacy.

H0: Perceived efficacy has no effect on self-efficacy.

Hypothesis 2: Perceived efficacy has a positive effect on response efficacy.

H0: Perceived efficacy has no effect on response efficacy.

Hypothesis 3: Perceived threat has a positive effect on perceived susceptibility.

H0: Perceived threat has no effect on perceived susceptibility.

Hypothesis 4: Perceived threat has a positive effect on severity.

H0: Perceived threat has no effect on severity.

Hypothesis 5: Repetition has a positive effect on response efficacy.

H0: Repetition has no effect on response efficacy.

Hypothesis 6a: Repetition has a positive effect on perceived susceptibility.

H6b: Repetition has a negative effect on perceived susceptibility.

H0: Repetition has no effect on perceived susceptibility.

Hypothesis 7: Perceived efficacy has a positive effect on behavior intention.

H0: Perceived efficacy has no effect on behavior intention.

Hypothesis 8a: Perceived threat has a positive effect on behavior intention.

H8b: Perceived threat has a negative effect on behavior intention.

H0: Perceived threat has no effect on behavior intention.

How many and which Conditions will Participants/Samples be Assigned to?

As mentioned in “How Many Observations will be Collected and What Rule will you Use to Terminate Data Collection?”, our planned maximum sample size is N = 602. There will be two conditions (i.e., being exposed to the message once meaning no-repetition condition and being exposed to the message twice meaning repetition condition). We will recruit 301 participants for each condition via Yahoo! Crowdsourcing (http://crowdsourcing.yahoo.co.jp/).

How many Observations will be Collected and what rule will you use to Terminate Data Collection?

We will perform structural equation modeling (SEM) with the data acquired from participants in the two conditions in the second wave. Based on a preliminary experiment, the minimum sample size for RMSEA-based SEM in the present experiment was found to be N = 301 in total, using the findRMSEAsamplesize function in R (MacCallum, Browne & Sugawara, 1996), with α = 0.05, power = 0.95, rmsea0 = 0.05, rmseaA = 0.01, df = 70. The degrees of freedom is calculated using the following formula: df = (1/2) {p (p + 1)} – q (Weston & Gore, 2006), where p is the number of observed variables (i.e., 14 in the model in Fig. 1), and q is the number of parameters to be estimated (i.e., 35 in the model). Given that there are two conditions in the present study, we set N = 151 as the required sample size for each condition.

We doubled the required sample size as the maximum sample size (i.e., N = 602 in total and 301 per condition) because we have to plan to collect data from a larger sample for the following two reasons: (a) A large amount of data may have to be excluded based on the criteria detailed below (i.e., “What are your Data Exclusion Criteria?”). (b) The dropout rate from the second wave may be high because it is hard to ensure all the participants in the first wave will take part in the second wave voluntarily within the requested time range, which is 24–72 h after the first wave.

If the collected data does not reach the required sample size (i.e., N = 301 in total and 151 per condition) after excluding the data meeting the criteria in “What are your Data Exclusion Criteria?”, we will collect data from additional participants to reach the required sample size. However, if the number of participants exceeds 602, we will choose the data of the first 602 participants based on the time stamp and use those for analysis.

What are Your Study Inclusion Criteria?

Participants will be recruited via Yahoo! Crowdsourcing (http://crowdsourcing.yahoo.co.jp/). They should possess Japanese nationality and be over 18 years old. We will indicate the criteria to potential participants in the instruction and invite them to participate only when they fulfill the criteria. Besides, questions on nationality and age will be asked before the main questionnaire (i.e., the items of the Risk Behavior Diagnosis Scale and behavior intention toward the target health proposal).

What are your Data Exclusion Criteria?

We will apply six criteria to perform data exclusion.To identify distracted respondents or satisficers (Chandler, Mueller & Paolacci, 2014; Oppenheimer, Meyvis & Davidenko, 2009; Sasaki & Yamada, 2019), we will insert a simple question as an attention check question (ACQ) in the middle of the questionnaire in the second wave. The reason why we do not add an ACQ in the first wave is that the first wave is conducted only to control the frequency of exposure to the message, and thus the data in the first wave will not be analyzed. The ACQ in the second wave will be: Please choose number “2” from below. Consistent with other items, the ACQ will also use a 7-point scale. The data from participants who choose 1 or 3–7 will be excluded.

To ensure that our manipulation on the frequency of exposure to the health proposal message is valid, the data from those who have seen this proposal before will be excluded. We will ask all participants the following question: “Have you seen this message before?” right after they are exposed to the message for the first time; we will ask the question in the first wave of the repetition condition and in the second wave of the no-repetition condition. The data from participants whose answer is “Yes” will be excluded.

For participants in the first wave of the repetition condition, a multiple-choice question concerning the content of the message (“What the message is about”) will be asked to confirm whether they read the message carefully enough to capture its meaning. Data from participants who give a false answer will be excluded.

Before analyzing the data, we will calculate the standard deviation (SD) of each participant’s scores on the 13 items in the second wave and exclude data from participants whose SD is zero. We plan to perform this data exclusion because an SD equal to zero means the same score on different items measuring divergent properties, which is strange and not possible.

As stated in “What are your Study Inclusion Criteria?”, data of participants whose nationality is not Japanese and whose age is under 18 will be excluded based on their answers to the relevant questions.

There will be an open-ended question on the experiment’s real purpose at the end of the questionnaire under both conditions in the second wave. The question will be optional, and the data from participants who give correct answers (consistent with the experiment’s real purpose) will be excluded.

What Positive Controls or Quality Checks will Confirm that the Obtained Results are able to Provide a Fair Test of the Stated Hypothesis?

Regarding the experiment design, to avoid unwanted interference from experiment materials (a health proposal message in our study), we conducted a pilot study to investigate whether it has been heard before and evaluated the agreement on the message. Considering that we want to discuss the effect of message repetition on the EPPM, a message already heard by most people is not appropriate. However, the COVID-19 pandemic’s impact is so broad that it is unlikely health proposals have not been heard. We can only try to find one that is known to relatively few people and exclude participants who have heard it before in the main experiment. Besides, a strongly supported message is prone to ceiling effects because there is no more space for improvement in the evaluation. As a result, we selected a message heard by 40.8% of the participants in the pilot study with an average agreement score of 5.35 (SD = 1.29, 7-point scale), which is not too high to induce ceiling effects. Moreover, our preliminary experiment also showed that there were no floor or ceiling effects (for more details, please refer to Supplemental 1).

During data collection and management, two questions will serve as manipulation checks on message repetition. The first question, which is for all participants when they are exposed to the message for the first time, is: “Have you seen this message before?” If the answer is “Yes,” it means there is interference from participants’ previous experience on our manipulation on message repetition. Thus, their data will be excluded. The other question, which will be asked only in the repetition condition in the first wave to ensure that the message’s first presentation is valid, is: “What is the message about?” Data from participants with false answers will be excluded.

Specify Exactly which Analyses you will Conduct to Examine the Main Question/Hypothesis(es)

Since SEM will be used to analyze the data, we will evaluate our model’s fit before proceeding to the hypotheses examination. It is reported that chi-square is sensitive to sample size (Schermelleh-Engel, Moosbrugger & Müller, 2003), and therefore, we will not rely on it as a basis for acceptance or rejection of the model. Instead, we will use root mean square error of approximation (RMSEA), comparative fit index (CFI), and Tucker–Lewis index (TLI) to evaluate the model’s fit. If RMSEA < 0.08, CFI > 0.9, and TLI > 0.9 (Kline, 2005; Bentler & Bonett, 1980) are all met, we will consider the model’s fit acceptable. If RMSEA < 0.06, CFI > 0.95, and TLI > 0.95 (Hu & Bentler, 1999) are all met, we will consider the model’s fit good. More details are described in Table 2.

Table 2 A design planner which specifies research questions, hypotheses, sampling plans, analysis plans, and contingent interpretation for “Specify exactly which analyses you will conduct to examine the main question/hypothesis(es)”.

Question	Hypothesis	Sampling plan (e.g., power analysis)	Analysis plan	Interpretation given different outcomes	
Does perceived efficacy underlie self-efficacy?	Hypothesis 1: Perceived efficacy has a positive effect on self-efficacy	As there are no other details to supplement, please refer to “How many observations will be collected and what rule will you use to terminate data collection?” for the sampling plan of the present study	We will analyze relevant indexes in the model using SEM. Specifically, we will use the false discovery rate (Benjamini & Hochberg, 1995) to adjust the p values of the coefficients of concern and then compare the adjusted p values with 0.05 to decide whether each coefficient is significant or not	For the health proposal in the present study, there is no evidence showing that its perceived efficacy underlies its self-efficacy in the Japanese context	
Does perceived efficacy underlie response efficacy?	Hypothesis 2: Perceived efficacy has a positive effect on response efficacy	For the health proposal in the present study, there is no evidence showing that its perceived efficacy underlies its response efficacy in the Japanese context	
Does perceived threat underlie perceived susceptibility?	Hypothesis 3: Perceived threat has a positive effect on perceived susceptibility			There is no evidence showing that the perceived threat of COVID-19 underlies its perceived susceptibility in the Japanese context	
Does perceived threat underlie severity?	Hypothesis 4: Perceived threat has a positive effect on severity			There is no evidence showing that the perceived threat of COVID-19 underlies its severity in the Japanese context	
Does repetition (increase in frequency of exposure to health proposal message from once to twice) influence response efficacy?	Hypothesis 5: Repetition has a positive effect on response efficacy			(1) The message is short while not interesting enough to elicit attention, thus, after 24–72 h, the memory of it may weaken and the effect becomes weak to detect
(2) The response efficacy is stable if the content of the health proposal message is fully understood from the beginning	
Does repetition (increase in frequency of exposure to health proposal message from once to twice) influence perceived susceptibility?	Hypothesis 6a: Repetition has a positive effect on perceived susceptibility
H6b: Repetition has a negative effect on perceived susceptibility			(1) The message is short while not interesting enough to elicit attention, thus, after 24–72 h, the memory of it may weaken and the effect becomes weak to detect
(2) Perceived susceptibility, as a kind of conjectural perception, is not directly connected with the content of the health proposal message, and thus may not be influenced by repetition	
Is behavior intention influenced by perceived efficacy?	Hypothesis 7: Perceived efficacy has a positive effect on behavior intention			In the COVID-19 pandemic, the perceived threat is so high that a fear control process is adopted, where perceived efficacy can be barely high enough to influence behavior intention	
Is behavior intention influence by perceived threat?	Hypothesis 8a: Perceived threat has a positive effect on behavior intention
H8b: Perceived threat has a negative effect on behavior intention			In previous research, the correlation between perceived threat and behavior indicators was not always significant (Witte, 1996), which means that the influence of perceived threat is weak. In the EPPM, perceived threat may act as a kind of judgment criterion, which means that the danger control process will not start until perceived threat reaches a certain level. After that, the fluctuation in its value does not influence behavior intention anymore	

Are you Proposing to Collect new Data or Analyze Existing Data?

The present study received approval from the psychological research ethics committee of the Faculty of Human-Environment Studies at Kyushu University (approval number: 2019-034). We will collect new data, and thus there will be no interference from existing data. This study consists of a series of anonymous online surveys. By participating in the surveys, participants consent to data collection.

Supplemental Information

Supplemental Information 1 A brief introduction of the preliminary experiment.

Click here for additional data file.

Additional Information and Declarations

Competing Interests

Author Contributions

Human Ethics

Data Availability

All authors declare there are no competing interests.

Jingwen Yang conceived and designed the experiments, performed the experiments, analyzed the data, prepared figures and/or tables, authored or reviewed drafts of the paper, and approved the final draft.

Xue Wu conceived and designed the experiments, performed the experiments, analyzed the data, prepared figures and/or tables, authored or reviewed drafts of the paper, and approved the final draft.

Kyoshiro Sasaki conceived and designed the experiments, performed the experiments, analyzed the data, prepared figures and/or tables, authored or reviewed drafts of the paper, and approved the final draft.

Yuki Yamada conceived and designed the experiments, performed the experiments, analyzed the data, prepared figures and/or tables, authored or reviewed drafts of the paper, and approved the final draft.

The following information was supplied relating to ethical approvals (i.e., approving body and any reference numbers):

The present study received approval from the psychological research ethics committee of the Faculty of Human-Environment Studies at Kyushu University (approval number: 2019-034).

The following information was supplied regarding data availability:

This is a protocol manuscript for Stage 1 registered report, and we have not conducted the experiment yet.

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
