# Peer review of "Changing health compliance through message repetition based on the extended parallel process model in the COVID-19 pandemic"

_PeerJ, doi:10.7717/peerj.10318_

## Round 0.1 · original submission · Major Revisions

Thank you for your submission. There are a number of issues with the submitted version that must be addressed. In particular, you need to refine your design section and add further clarity with regard to your research aims.

·

Basic reporting

• Line 2 - 4: Your abstract needs modification. How you relate fear with health messages is not clear. Health messages are not necessarily be fearful I suggest that you improve the description at lines 2-4 to provide more description for rationale.
• Line 5 - 6: The title doesn't tell about the first aim (“EPPM in naturalistic context…”)...rather it focuses only on the second objective.
• You described the association between health proposal on Susceptibility. However, it is not clear what kind of susceptibility you referring for, physiological or perceived. I think you mean perceived susceptibility. Be explicit in description.
• Your description of the rationale in Line 35-36, you mentioned as there are inconsistent findings. It needs more elaboration on the inconsistent findings by providing necessary studies.
• In line 84-85 of the introduction, you mentioned that “Even though the degree to which one is influenced by the pandemic may vary, the cognition of its perceived threat can be developed spontaneously.” However, the cognition of perceived threat also vary to a large extent as the influence varies.
• The way you write the hypothesis needs modification. It looks, in any way the hypothesis is true regardless of the data.
I suggest to formulate it like;
H1: Perceived efficacy has a positive/negative/ (depending on whether one- or two-sided test) effect on self-efficacy.
H0: Perceived efficacy has no effect on self-efficacy.
• Table: you frequently mentioned “Based on a preliminary experiment….”. Could you cite the results of the preliminary experiment or upload as supplementary material?
• The interpretation column of the table need to be revised. The one written here is a general rule not specific to your study and context. It is also a redundancy from the hypothesis section.
• The table in the analysis plan is not relevant. All are redundancy from the hypothesis section and sampling. You can summarize it using a single paragraph or simple table provided that there is no specific analysis for each hypothesis.

Experimental design

• In your research question, line 28-29 you described “to examine how people health compliance intention is influenced by various factors…….”. This research question is not reflected in the title. Your title needs modification to accommodate most of the research aims.
• Line 12-13: The study aimed to compare one vs two-time messages. However, the description in line 12-13 (‘the times of exposure’) and also in the main manuscript looks as you have multiple exposure times. It is not possible to assess the association of the ‘times’ of exposure with response efficacy. The difference between one vs two messages might not be necessarily equal to the difference in two vs three messages. Your interpretation seems continuous exposure variable but it is binary.
• In this study, a number of hypotheses are being tested. Multiple comparison is still an issue. As you are evaluating the statistical significance of multiple parameters in a model, the type I error is expected to inflate. You should describe how to control the multiplicity issues. What type of statistical adjustment will you use?
• In line 197-199 you described how you determine sample size. However, the sample size calculation is not clear. The ‘findRMSEAsamplesize’ function of R requires a degree of freedom to calculate sample size and the sample size varies accordingly. What degree of freedom did you used? How much is the expected dropout rate and you need to adjust for it as well.
• In you inclusion criteria, in line 222-224: Will you use the nationality as screening or they will fill out the questionnaire regardless of their nationality?
• The data exclusion and sample size adjustment is not clear. Needs more elaboration.
• In line 246-247 of the data exclusion section you mentioned that participants with SD of 0 will be excluded. I didn’t get the reason to exclude the participants with SD of 0. The measurement is only twice, it is highly possible to have same score i.e. SD of 0. Why you exclude them?
• The quality check and control needs descriptions in detail what quality checks will be performed during the design, data collection, and management.
Analysis plan
• It is recommended to use Tucker–Lewis index in addition to RMSEA and CFI to assess the model fit.
• Page 24: The items looks similar. Any validity test for these items? How is the inter-item correlation of the questionnaire? In general, the psychometric validity of all the tools to be used need to be described in detail.

Validity of the findings

• As a protocol, I couldn’t assess the validity of the findings comparing with the data.
• You mentioned about the preliminary experiment you conducted, however, neither the data nor the results are provided.

Additional comments

• Line 11-12: It says “SEM is used to....” As it is a protocol, it needs to be corrected to “SEM will be used….”
• You repeatedly used the phrase “we predicted” for instance line 12, 15… You mean hypothesized? These two terms are different.
• Line 108-111: Put the direct translation of the message in English.
• Line 254-256: This sentence should be part of the exclusion not the quality control
• The description written in the “Sampling plan” column of the Table is repetition. You should remove it.
• RMSEA, CFI – Put the long form at first instance - root mean square error of approximation - RMSEA, comparative fit index - CFI.

Reviewer 2 ·

Basic reporting

Yang and colleagues seek to probe factors relating to the EPPM model of fear-motivated health behaviors as well as frequency of measure exposure that may affect behavior intention in the context of the Covid-19 pandemic. Such a study may identify features that have led to relative success in managing the pandemic in some countries relative to others. The proposed study would certainly be beneficial in our attempt to understand and combat the pandemic. Further, the proposed study benefits from a clean and elegant experiment, that can be easily understood by respondents and then further operationalized for analysis. One omission is of preliminary analyses of the pilot data. The study also does not appear to involve any neuroscience and therefore should not be listed as such .

Experimental design

• 2. Experimental design

1. I recommend changing the ACQ to something less subjective. You risk biasing your sample towards those that already formulate their beliefs based on evidence, which does not comprise the general population, or at the very least, lose participants who find it humorous to answer > 1. Perhaps a simple request to select a number on the scale could serve as an ACQ.
2. There is also a concern regarding the nationality restriction, given that the question regards a global pandemic. There may be IRB limits to your participant pool but it may be advantageous to expand the survey population, if possible, to other nationalities.
3. 1. In line 23, the message to be repeated is first assessed according to favorability but it is unclear what is meant by this measure. Is it assessing how much respondents: 1) like the message; 2) agree with the message?

Validity of the findings

N/A

Additional comments

Major comments

It is a bit unclear what the specific aim /question of the study is. In the abstract, I can identify 4: response efficacy ~ frequency of exposure; susceptibility ~ frequency of exposure; behavior intention ~ perceived efficacy; behavior intention ~ perceived threat.
The stated aims in the rest of the report are fuzzier. There are some attempts, in lines 77-78; 96-98; 101-103, but they are vague. This part of the report needs to be tightened.
With regards to the specific hypotheses listed, there are too many and some may not be pertinent, notably hypotheses 1-4.

Perhaps providing a real-world translation in the interpretation column could help. But it is not clear why finding a relationship between perceived and self-efficacy is of interest or utility in the context of the proposed study. The same criticism underlies hypothesis 2. With hypotheses 3 and 4, one can argue that it is highly likely that perceived threat (i.e. some average value of susceptibility and severity) is related to susceptibility and severity.


Minor comments:

1. In the opening paragraph of the introduction, you use colloquial and vague language (“It has been a while…”). Please be more specific, so as to allow the reader to contextualize the problem (e.g. “As of writing, it has been x months since the outbreak (…)”. Second, the impact on life and financial well-being are not incalculable, as both life and economic loss can be counted.

2. Some language throughout the report is a bit colloquial (e.g. “sum up” instead of “in summary) and there are a few typos (Hypothesis 4 is listed twice in the table). A revision of the writing would benefit the report.

3. Please change “times of exposure” to “frequency of exposure” to avoid confusion.

Reviewer 3 ·

Basic reporting

The project meets basic report standards. The table is somewhat large and difficult to read, however, as the cells aren't always aligned well. (eg., sampling plan doesn't track all the way down with the other cells)

There are a few minor English errors in the manuscript and it should be thoroughly proofread.

Experimental design

The experimental design is poor. The primary manipulation isn't very strong (1 vs 2 repetitions) and there is no control group. Most of the hypotheses are regarding structural relationships that are well-established and unlikely to be impacted by repetition very extensively. There aren't safe guards to ensure that any changes aren't due to demand characteristics.

Validity of the findings

I doubt the current study will lead to robust repetition effects, and it is lacking evidence that it should.

---

## Round 0.2 · Minor Revisions

Thank you for your revised version. The reviewers agree that this is much improved, however there are a few issues to address.

·

Basic reporting

• I still have some concerns on the description in line 96-103. The declaration of WHO as pandemic alone could not be considered as the health treat by all the people to the same extent. The way people react to the this health treat (COVID-19) considerably vary according to the type of health messages communicated by each country and even specific contexts. For instance, the health messages in Belgium are different from the messages in Uganda or Ethiopia, which in turn leads variation in the influence on peoples. I agree that the government measures could be the same as WHO standard. However, the health messages and respective reactions significantly varies within and across countries.
• The formulation of the hypothesis is improved. I still have minor issues. The H1 is one sided while the Ho is two sided. For instance, if perceived efficacy has a negative effect on self-efficacy, what conclusions would be made? This is the same for some of the hypotheses.

Experimental design

• Good to add the degree of freedom. However, not clear why select 70 as df? Since it is a protocol, it is better if you add a description to justify it.
• I suggest to put age as relevant characteristics as inclusion criteria.

Validity of the findings

Not applicable

Additional comments

Thank you for the opportunity to review this manuscript once again. Overall, the manuscript has significantly improved. The preliminary experiment substantiates the need for the present study. I have a few minor comments.

Reviewer 2 ·

Basic reporting

n/a

Experimental design

Overview: Thank you for revising your report. I still maintain that H1-4 are not true hypotheses, as the outcome variables are constructed from the dependent variables. Related coefficients are not included in supplementary 1, Figure 1’. I leave it to the authors to consider when submitting their results.

Some examples of this relationship:
Abstract: Here, you claim that “perceived threat of COVID-19, which underlies perceived susceptibility”, so why test this in H4?

Lines 64-65: If perceived efficacy is composed of self-efficacy and response efficacy, I still don’t understand why their relationships is being tested in Hypotheses 1 & 2. You could use these hypotheses as positive controls however.

Validity of the findings

n/a

Additional comments

Line 187: While the term “ social distancing” has gained ground, it is at base incorrect. Please change the term to “physical distancing”.

---

## Round 0.3 · accepted · Accept

Thank you for your revised article. I am pleased to inform you that this has now been accepted for publication - congratulations!